# Exploring Perceptions and Barriers: A Health Belief Model-Based Analysis of Seasonal Influenza Vaccination among High-Risk Healthcare Workers in China

**DOI:** 10.3390/vaccines12070796

**Published:** 2024-07-18

**Authors:** Yalan Peng, Yantong Wang, Wenzhi Huang, Ji Lin, Qinghui Zeng, Yi Chen, Fu Qiao

**Affiliations:** 1Center of Infectious Diseases, West China Hospital, Sichuan University, Chengdu 610041, China; yalanpeng@wchscu.edu.cn; 2Department of Infection Control, West China Hospital, Sichuan University, Chengdu 610041, China; wangyantong@wchscu.cn (Y.W.); huangwenzhi@wchscu.cn (W.H.); linji@scu.edu.cn (J.L.); zengqinghui@wchscu.edu.cn (Q.Z.); toddy@scu.edu.cn (Y.C.)

**Keywords:** seasonal influenza vaccination, healthcare workers, Health Belief Model, vaccination acceptance

## Abstract

The annual seasonal influenza vaccination rate among high-risk healthcare workers (HCWs) has fallen below expectations, underscoring the importance of exploring the impact of perception on vaccination behavior. An online survey, grounded in the Health Belief Model (HBM), was administered to high-risk healthcare workers at West China Hospital. The data analysis encompassed descriptive statistics, logistic regression for univariate analysis, and path regression for multivariate analysis. A total of 1845 healthcare workers completed the survey, with an acceptance rate of 83.90% (95% CI, 82.20–85.60%). Path analysis revealed significant correlations between vaccination acceptance and perceived susceptibility (β = 0.142), perceived benefits (β = 0.129), perceived barriers (β = 0.075), exposure to vaccination advertisements (β = 0.115), and knowledge about seasonal influenza (β = 0.051). Vaccination education efforts should prioritize elucidating the risks associated with the disease and emphasizing the benefits of vaccination. Furthermore, leveraging advertising proves to be an effective strategy for promotion.

## 1. Introduction

Healthcare workers (HCWs) are at high risk of seasonal influenza. A meta-analysis by Kuster et al. retrieved relevant literature from 1947 to 2020 worldwide, covering 13,373 participating healthcare workers, revealing that the incidence of seasonal influenza among unvaccinated healthcare workers was 18.57%, which was 3.4 times higher than among general adults [1]. Vaccination has already proved to be an effective way to prevent seasonal influenza. However, a meta-analysis by Imai retrieved global literature related to influenza vaccination among healthcare workers from 1980 to 2018, indicating that the vaccination rate among healthcare workers worldwide ranged from 16.9% to 69.4%, which is lower than what is generally expected [2].

The reluctance of healthcare workers to accept seasonal influenza vaccination has been examined in previous studies, which have identified several reasons, including low-risk perception [3], fear of adverse reactions [4], and lack of vaccine confidence [5]. The Health Belief Model has been used to explore influencing factors of seasonal influenza vaccine uptake, as a structured model [6].

However, different studies have yielded different conclusions. For instance, a multicenter study conducted in Brunei, Hong Kong, and Singapore found that the influencing factors were perceived susceptibility, perceived benefits, and cues to action [7]. Conversely, a study conducted in Jordan only identified the perceived benefits as a significant factor [8]. These discrepancies may stem from cultural disparities across different countries [9]. Additionally, attitudes toward vaccines are not solely individual decisions but are also influenced by social belonging and identity [10].

In our hospital, the vaccination rate was only 59.22% during the 2022 flu season, despite the implementation of a free, on-site vaccine strategy. However, to date, no research has investigated the reasons for healthcare workers in mainland China being unwilling to receive influenza vaccines based on the Health Belief Model. Therefore, we conducted this study to identify effective measures to improve the acceptance of seasonal influenza vaccines among healthcare workers.

## 2. Materials and Methods

### 2.1. Study Design and Setting

A cross-sectional survey was conducted among healthcare workers in high-risk wards at West China Hospital of Sichuan University from 4–30 November 2022. The high-risk wards were identified as those where the number of seasonal influenza cases ranked among the top ten in the hospital, or where there was a cluster of seasonal influenza cases in 2021. The healthcare workers encompassed all medical-related staff, including physicians, surgeons, nurses, technicians, ancillary workers, researchers, and administrators. An online questionnaire survey was administered to the healthcare workers using the Wenjuanxing online survey system, provided by Ranxing Information Technology Co., Ltd., Changsha, China. The questionnaire was distributed through the hospital’s official WeChat to the work groups of each high-risk department, where the WeChat group included all employees of the department, ensuring that all potential respondents received the questionnaire. Permission was obtained from the department heads before distributing. To ensure a high response rate and to raise awareness among department staff, we sent the questionnaire to the infection control nurses of each department, who then distributed it to the official WeChat group of that department. The department heads further issued a notice reminding all staff to fill out the questionnaire, regardless of their intention to receive an influenza vaccination. Additionally, the infection control nurses reminded them every day. For those healthcare workers who were unable to complete the online questionnaire, the infection control nurses were responsible for organizing the completion of a paper-based version and manually entering the data. The completion of the questionnaire was based on voluntary participation without consent forms. This study was approved by the Ethics Committee on Biomedical Research, West China Hospital of Sichuan University (Project Number: 13872023).

### 2.2. Survey Questions

The questionnaire was developed by two infection control practitioners and reviewed by three infection control experts, and was structured around the Health Belief Model. The questionnaire was developed based on existing validated measures that used the Health Belief Model [11]. After reviewing the relevant literature, in addition to basic demographic information and influenza knowledge mastery, we have also included all the dimensions of the Health Belief Model [12,13,14]. Meanwhile, each dimension of the Health Belief Model has been structured into questions from three perspectives: protecting oneself; protecting family members; and protecting patients [15]. Responses in the Health Belief Model sections used a three-point Likert scale: (“agree” or “very concerned” = 3; “not sure” or “moderately concerned” = 2; and “disagree” or “not concerned” = 1). Inverse items were reverse-scored. It consisted of four parts, in total 31 questions. Part one collected sociodemographic data. Part two focused on the primary outcome of the survey (acceptance of the seasonal influenza vaccine). Participants were asked, “Are you planning to receive the flu vaccine as the hospital is organizing an on-site and free influenza vaccination campaign?” with response options “Yes” or “No”. Part three assessed knowledge factors. Part four explored the constructs of the Health Belief Model, which included five constructs according to the presented hypothesis: perceived severity; perceived susceptibility; perceived barriers; perceived benefits; and cues to action. Each construct had three to four questions, making a total of 16 questions for the Health Belief Model section.

### 2.3. Data Management and Analysis

The data were exported into SPSS version 21.0 (IBM, New York, NY, USA) from the online survey platform. A power analysis was conducted to determine the number of participants needed in this study, using R 4.2.0 (Lucent, New Jersey, NJ, USA). Any questionnaire responses completed within 30 s were discarded, as they indicated a lack of attention, based on the shortest time of completing the questionnaire from our small-scale pilot experiments before the investigation [16]. Inconsistent answers (e.g., selecting “Yes” for being willing to be vaccinated at their own expense, but “No” for intending to be vaccinated) were also discarded. Responses that did not align with the actual situation were deleted, such as those indicating an age of less than 25 years but with a senior professional title. A validity and reliability test had been conducted. Cronbach’s alpha was used to calculate the internal consistency reliability of each construct [17]. Factor analysis, employing varimax rotation, was conducted to validate the questionnaires. Descriptive analysis was conducted to demonstrate the impact of sociodemographic factors, knowledge factors, and the Health Belief Model on acceptance. Logistic regression was employed to estimate the relationship between factors and acceptance. For each factor, odds ratios (ORs) and 95% confidence intervals (CIs) were calculated to quantify its influence. Path analysis was conducted using AMOS version 24.0 (IBM, New York, NY, USA) to control potential confounders and examine the direct and indirect effects of multiple variables on acceptance. The model fit was assessed by goodness of fit (GFI), adjusted GFI (AGFI), root mean square residual (RMR), the root mean square error of approximation (RMSEA), comparative fit index (CFI), normed fit index (NFI), non-normed fit index (TLI), and the maximum likelihood chi-square values/degrees of freedom ratio [18]. A *p*-value < 0.05 was considered significant.

## 3. Results

### 3.1. Study Population

The response rate of the questionnaire was 65.80% (1845/2804). A total of 2804 healthcare workers across all 23 high-risk wards participated in the survey, with 1848 of them responding. Out of the 1848 questionnaires, 1845 passed the quality assessment and were included in the analysis, which is significantly higher than the expected sample size of 1033. Among the study group, 85.96% (*n* = 1586) were female and 14.04% (*n* = 259) were male. Most of them were aged between 25 and 50 years (*n* = 1324, 71.76%). The majority of participants belonged to three occupation groups: nurses (*n* = 1117, 60.54%); physicians (*n* = 348, 18.86%); and technicians (*n* = 148, 8.02%). Only 8.84% (*n* = 163) of the participants held associate senior or senior professional titles. Regarding knowledge factors, 66.07% (*n* = 1219) of the participants agreed that they were familiar with influenza vaccine-related knowledge, and 66.45% (*n* = 1226) of them agreed that they were familiar with influenza-related knowledge. The acceptance rate of seasonal influenza vaccination among healthcare workers in these high-risk wards was 83.90% (95% CI, 82.20–85.60%).

In the logistic analysis, among sociodemographic characteristics, age, and professional title were significant influencing factors for the acceptance of the influenza vaccine. Healthcare workers over 50 years old were more likely to receive the influenza vaccine, compared with those aged under 25 (OR = 0.028) or aged 25–50 (OR = 0.040). Healthcare workers with associate senior (OR = 3.336) or middle (OR = 1.754) professional titles were more likely to receive the influenza vaccine, compared with unclassified staff. For occupation, surgeons (OR = 0.32) were less likely to accept the influenza vaccine compared with physicians. Among the knowledge score factors, knowledge score on seasonal influenza vaccination (OR_1_: moderate vs high = 0.34, OR_2_: low vs high = 0.21) and knowledge score on seasonal influenza (OR_1_: moderate vs high = 0.37, OR_2_: low vs high = 0.21) had a significant impact on the acceptance of the influenza vaccine. This is shown in Table 1.

### 3.2. Reliability Analysis and Validity Analysis

All factors achieved an internal reliability of alpha > 0.6. The KMO test (KMO = 0.841) and Bartlett’s Test of Sphericity (Approx.Chi-square = 32,681.572, *p* < 0.05) confirmed the adequacy of exploratory factor analysis. The factor analysis resulted in five constructs, similar to the Health Belief Model hypothesis.

### 3.3. Univariate Analysis by Logistic Regression

All items related to perceived severity, perceived susceptibility, and perceived benefit have a significant impact on healthcare workers’ acceptance of receiving vaccines. Among perceived barriers to vaccination, fear of adverse reactions or needles can hinder healthcare workers’ willingness to vaccinate (*p* < 0.05), but the availability of time does not affect vaccination acceptance. Among the cues to action, leader initiatives (*p* < 0.05), or vaccination advertisements (*p* < 0.05) can promote vaccination acceptance, but family supervision has no significant effect on promoting vaccination acceptance. The five constructs within the Health Belief Model have a significant impact on healthcare workers acceptance of receiving vaccines, as shown in Table 2.

### 3.4. Multiple Factor Analysis by Path Analysis

The path analysis revealed direct and indirect effects of acceptance of seasonal influenza vaccination. It demonstrated a positive direct effect of acceptance of seasonal influenza vaccination via perceived susceptibility (β = 0.142; 95% CI, 0.106–0.174), perceived benefits (β = 0.129; 95% CI, 0.078–0.189), perceived barriers (β = 0.075; 95% CI, 0.045–0.100), knowledge score on seasonal influenza (β = 0.051; 95% CI, 0.018–0.083), as well as advertisements promoting seasonal influenza jabs within the hospital (β = 0.115; 95% CI, 0.069–0.161) (Figure 1a). Indirect effects also exist. Advertisements for seasonal influenza jabs in the hospital have an indirect effect, which is mediated positively by perceived susceptibility (β = 0.247; 95% CI, 0.184–0.305) and perceived benefits (β = 0.221; 95% CI, 0.171–0.276). Knowledge about seasonal influenza has an indirect effect, which is mediated positively by perceived susceptibility (β = 0.139; 95% CI, 0.094–0.185), perceived barriers (β = 0.157; 95% CI, 0.110–0.201), and perceived benefits (β = 0.172; 95% CI, 0.136–0.208) (Figure 1b). The results showed that the path analysis had comparable model fit indices (CMIN/DF = 2.527, GFI = 0.999, AGFI = 0.990, RMR = 0.002, RMSEA = 0.029, CFI = 0.997, NFI = 0.996, TLI = 0.980).

## 4. Discussion

Our study found that the acceptance rate of seasonal influenza vaccine among healthcare workers was 83.90% (95% CI, 82.20–85.60%). The vaccine acceptance rate among healthcare workers reported in two meta-analyses published in 2021 and 2023 ranged from 16.8% to 61%, representing a wide span [19,20]. However, the vaccine acceptance rate investigated in this study is significantly higher than those figures. The acceptance rates varied among different countries. For instance, in 2011, a study in the UK reported that 93.8% of healthcare workers received seasonal influenza vaccines [21], whereas in 2019, a study in Germany reported that only 38.8% of physicians were prone to receive the vaccines [22]. A meta-analysis in 2021 retrieved globally published literature on healthcare workers’ vaccine acceptance from 2000 to 2019 and quantitatively demonstrated significant differences in vaccine acceptance among healthcare workers from different continents. Specifically, the acceptance rate of influenza vaccines among Asian healthcare workers was 69%, significantly higher than that in Europe, which stood at 54% [20]. This also explains why it is not feasible for us to directly adopt the research results from other countries investigating the influencing factors of healthcare workers’ influenza vaccine acceptance. In 2017, a study in China showed that the influenza vaccination rate among healthcare workers was 5%, while the vaccination rate among the elderly aged 60 and above was 0.8%, and the rate among pregnant women was 4% [23]. The higher vaccination rate among healthcare workers is due to their possession of more knowledge about influenza and influenza vaccines, as well as the provision of free vaccination services by some hospitals for healthcare workers.

In this study, the logistic analysis revealed that acceptance of seasonal influenza vaccination increased with age, title and knowledge factors. First, in terms of age, this study posits that age is a contributing factor to vaccination. Although age has been reported to have either a positive or negative impact on vaccine acceptance in different studies [19,24], our research findings are reasonable within the cultural context of China. In China, older people tend to be more concerned about their own health, leading to a greater understanding of vaccines and, consequently, higher acceptance rates. This finding aligns with a trend observed in a study conducted in Oman in 2020 [25]. Nevertheless, it is worth noting that the survey results of this study indicate that the complete awareness rate among healthcare workers, regarding both influenza and influenza vaccines, is only 66%, which is far below the expected level. Occupation was also an influential factor. In studies conducted in 2012 and 2023, contrasting results were presented regarding whether doctors or nurses were more inclined to receive vaccinations [16,24]. In our study, we further subdivided doctors into surgeons and physicians and found no significant difference in the acceptance rate between physicians and nurses. However, surgeons had a significantly lower acceptance rate. The trend observed in other studies that gender influenced acceptance was not seen in our study [16,24,26]. Additionally, we observed that healthcare workers showed the same level of concern for issues related to protecting themselves, their families, and their patients across all perceived dimensions. This finding is consistent with the conclusion of the meta-analysis published in 2014 [15]. Based on these findings, we recommend adopting stratified training measures for healthcare workers. For younger healthcare workers, it is necessary to capture their attention through more diverse new media promotion channels. To address the lower acceptance rate of influenza vaccines among surgeons, measures can be taken to draw their attention through the presentation of influenza-related data, such as adverse event rates, effectiveness rates, or case studies, and by popularizing influenza vaccine knowledge. Currently, existing influenza vaccine promotion strategies tend to emphasize the protective effect on healthcare workers themselves, often neglecting the equally important role of protecting their families and patients. The conclusion of this study suggests that more emphasis should be placed on explaining the benefits of protecting families and patients in the promotion efforts.

In this study, path analysis disclosed how sociodemographic characteristics, knowledge factors, and constructs of the Health Belief Model influence acceptance. According to the analysis, perceived susceptibility (β = 0.142), perceived barriers (β = 0.075), perceived benefits (β = 0.129), advertisement in hospital (β = 0.115), and knowledge about seasonal influenza (β = 0.051) showed significant associations with acceptance. Three Asian studies, originating from Israel (2017), Jordan (2020), and Singapore (2019) have converged on a shared insight: perceived benefits significantly influence influenza vaccine acceptance among healthcare workers [8,27,28]. Additionally, the Singapore study also identified perceived severity and perceived barriers as influencing factors. Three meta-analyses presented different results. Two of these meta-analyses, published in 2012 and 2014 respectively, concurred that perceived susceptibility and perceived benefits were influencing factors [15,24], while the one published in 2023 considered perceived benefits, perceived severity, and perceived barriers as influencing factors [19]. These findings differ from the results of our study. Research has revealed that cultural specificities play a crucial role in vaccination acceptance [9]. Based on the results of this study, healthcare workers have specific preferences for information promoting influenza vaccination. The promotional information should focus on the transmission routes and susceptibility of influenza, using data to illustrate the adverse reactions and protective effects of receiving the influenza vaccine. To eliminate obstacles to influenza vaccination, it is important to address the issue of healthcare workers’ availability during the regular working hours of vaccination centers, as this can interfere with their primary duties. Extending vaccination hours, especially non-standard hours, will accommodate healthcare workers and increase their willingness to get vaccinated. The selection of vaccination sites should prioritize locations convenient for healthcare workers, or adopt mobile vaccination services, to increase the actual vaccination rate. Additionally, promotional videos featuring vaccinated healthcare workers sharing their personal experiences can help alleviate fears about needles. It is noteworthy that the commonly adopted approach of department heads urging healthcare workers to get vaccinated has been shown to be ineffective. In-hospital advertisements and promotions are more acceptable forms of advocacy for healthcare workers. This indicates that using data and case studies to illustrate the significance and safety of influenza vaccination is crucial, encouraging healthcare workers to voluntarily choose vaccination rather than relying on administrative measures to mandate it.

It is notable that not all constructs of the Health Belief Model were highly correlated with acceptance. This suggests that there may be other factors influencing healthcare workers’ acceptance that are not encompassed within the Health Belief Model, warranting exploration in future research.

## 5. Limitation

This study presents several limitations. First, it was conducted within a single institution; however, it involved 1845 healthcare workers in 23 wards. Therefore, we contend that our findings are generalizable to other general hospitals in mainland China. Furthermore, the factors influencing the willingness of staff in other medical institutions in China to receive influenza vaccination, such as geriatric hospitals, long-term care facilities, community hospitals, and so on, will also be a topic of further attention in the future. Given the different staff compositions and institutional cultures across various types of institutions, there may be variations in strategies to promote vaccination. Second, the response rate was only 65.80%, which is not uncommon in similar studies. For instance, Wong’s research in three Southeast Asian countries reported response rates ranging from 11% to 36% [29], while Polla’s research yielded a response rate of 61.2% [30], similar to this study. Third, it was a cross-over study, and a random sampling method was not utilized to collect data; instead, the questionnaire was disseminated to all healthcare workers in high-risk wards on a voluntary basis, potentially introducing selection bias. At the same time, we compared the demographic information of the surveyed and non-surveyed groups among our research subjects. At the position level, significant differences do exist. However, in the univariate analysis, only the acceptance rate among surgeons showed significant differences compared to other positions, but the number of this group is extremely small, having minimal impact on the final results of the path analysis. To mitigate this bias, we obtained consent from ward leaders before conducting the investigation, and they assisted in encouraging staff to complete the questionnaire, regardless of whether they had received the vaccine, which increased the response rate and eliminated selection bias to some extent. Fourth, the questionnaire utilized in this study was self-designed, grounded on the Health Belief Model and informed by research from other pertinent literature. This was necessitated by our need to convey the content in Chinese and to specifically delve into the preferences of healthcare workers regarding “self-protection, family protection, and patient protection”. Our self-crafted questionnaire has undergone rigorous reliability and validity tests, hence our decision not to utilize widely established and validated scales. Lastly, this study, based on the response duration from the pre-experiment, retained questionnaires with a response time exceeding 30 s, which is shorter than the conventional threshold; however, we have ensured that these questionnaires meet other quality control criteria.

## 6. Conclusions

The results of our study suggest that the factors influencing influenza vaccination rates among healthcare workers include enhancing the perception of susceptibility and the benefits of vaccination. Emphasizing influenza severity does not increase acceptance. Chinese healthcare workers display equal concern for self, family, and patient protection, which should be emphasized in promotions. Reforms in promotion and education content are crucial, focusing on healthcare workers’ concerns to encourage vaccine acceptance.

Healthcare workers’ knowledge about influenza and vaccines requires improvement. Only 66% demonstrated a high level of knowledge. Knowledge was a factor promoting acceptance, but assumptions of high knowledge contributed to low acceptance rates.

Unconventional vaccination times and convenient locations or mobile methods cater to healthcare workers’ schedules. Traditional approaches of department heads appealing for vaccination have limited effectiveness. Instead, promoting subjectivity and highlighting the significance of vaccination are important. Tailored multi-channel approaches, including data and case studies, can address lower acceptance among younger healthcare workers.

## Figures and Tables

**Figure 1 vaccines-12-00796-f001:**
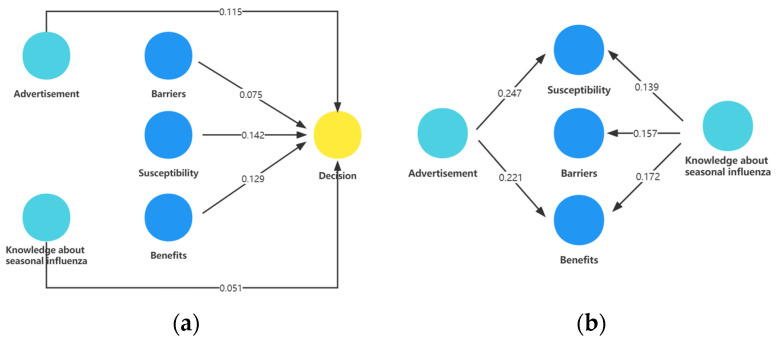
The path model of constructs in relation to acceptance of seasonal influenza vaccination. The yellow circle represents the acceptance of seasonal influenza vaccination, the blue circle represents perceived dimensions of the Health Belief Model, the green circle represents knowledge factors and factors of cues to action. Data on the line represent influence coefficient: (**a**) Direct effect of acceptance of seasonal influenza; (**b**) Indirect effect of acceptance of seasonal influenza mediated by perceived susceptibility, perceived barriers, and perceived benefits.

**Table 1 vaccines-12-00796-t001:** The acceptance of seasonal influenza vaccination among healthcare workers varies according to sociodemographic characteristics and knowledge factors (*n* = 1845).

Characteristics	Number(% ^a^)	Acceptance of Seasonal Influenza Vaccine		
		*n*(% ^b^)	OR	95% CI
**Sociodemographic characteristics**				
Age group				
<25	389(21.08%)	305(78.41%)	0.028 *	0.004–0.201
25–50	1324(71.76%)	1112(83.99%)	0.040 *	0.006–0.288
≥50	132(7.16%)	131(99.24%)	Ref	-
Gender				
Male	259(14.04%)	225(86.87%)	1.316	0.896–1.932
Female	1586(85.96%)	1323(83.42%)	Ref	-
Occupation				
Surgeon	16(0.87%)	10(62.50%)	0.32 *	0.11–0.97
Nurse	1117(60.54%)	953(85.32%)	1.11	0.80–1.54
Technician	148(8.02%)	118(79.73%)	0.75	0.46–1.25
Administrator	39(2.11%)	32(82.05%)	0.88	0.39–2.25
Ancillary workers	104(5.64%)	85(81.73%)	0.86	0.50–1.55
Others **	73(3.96%)	58(79.45%)	0.74	0.40–1.44
Physician	348(18.86%)	292(83.91%)	Ref	-
Professional Title				
Senior	56(3.04%)	56(100%)	-	-
Associate senior	107(5.80%)	100(93.46%)	3.336 *	1.503–7.401
Middle	366(19.84%)	323(88.25%)	1.754 *	1.192–2.580
Junior	793(42.98%)	645(81.34%)	1.018	0.767–1.350
Unclassified	523(28.35%)	424(81.07%)	Ref	-
**Knowledge factors**				
Knowledge score on seasonal influenza vaccination				
Low (Score 1)	76(4.12%)	49(64.47%)	0.213 *	0.129–0.353
Moderate (Score 2)	550(29.81%)	408(74.18%)	0.337 *	0.259–0.439
High (Score 3)	1219(66.07%)	1091(89.50%)	Ref	-
Knowledge score on seasonal influenza				
Low (Score 1)	64(3.47%)	40(62.50%)	0.205 *	0.120–0.350
Moderate (Score 2)	555(30.08%)	416(74.95%)	0.367 *	0.282–0.478
High (Score 3)	1226(66.45%)	1092(89.07%)	Ref	-

^*^ *p*-value < 0.05. ^**^ others include researchers, medical support staff. ^a^ refers to composition ratio, which is the proportion of each category in the overall population. ^b^ refers to the proportion of individuals in each category who accept the influenza vaccine.

**Table 2 vaccines-12-00796-t002:** Acceptance of the seasonal influenza vaccine by the Health Belief Model (*n* = 1845).

HBM Constructs	Acceptance of Seasonal Influenza Vaccine			
	Accept (%)	Not Accept (%)	OR	95% CI
**C1. Perceived severity of infection**				
Q1. How severe the seasonal influenza would be if you got it?				
Low	237(79.26%)	62(20.74%)	0.429 *	0.303–0.607
Moderate	443(76.64%)	135(23.36%)	0.374 *	0.282–0.497
High	868(89.76%)	99(10.23%)	Ref	-
Q2. In what extent the seasonal influenza would affect your family member if you got it?				
Low	215(78.18%)	61(21.82%)	0.396 *	0.279–0.562
Moderate	383(74.37%)	132(25.63%)	0.321 *	0.242–0.425
High	950(90.05%)	105(9.95%)	Ref	-
Q3. In what extent the seasonal influenza would affect your work if you got it?				
Low	188(77.37%)	55(22.63%)	0.412 *	0.289–0.587
Moderate	339(74.02%)	119(25.98%)	0.343 *	0.259–0.454
High	1021(89.25%)	123(10.75%)	Ref	-
**C2. Perceived susceptibility to infection**				
Q4. How likely do you think you got seasonal influenza?				
Low	123(56.68%)	94(43.32%)	0.118 *	0.082–0.168
Moderate	579(82.01%)	127(17.99%)	0.410 *	0.302–0.555
High	846(91.76%)	76(8.24%)	Ref	-
Q5. Do you think HCW are more likely to get seasonal influenza than general population?				
Low	100(63.69%)	57(36.31%)	0.216 *	0.150–0.313
Moderate	272(76.19%)	85(23.81%)	0.353 *	0.264–0.472
High	1176(89.02%)	145(10.98%)	Ref	-
Q6. How likely do you think your family member would catch seasonal influenza and infect you?				
Low	131(61.79%)	81(38.21%)	0.171 *	0.122–0.240
Moderate	385(78.25%)	107(21.75%)	0.380 *	0.284–0.509
High	1032(90.45%)	109(9.55%)	Ref	-
**C3. Perceived barriers against vaccination**				
Q7. I am afraid of getting adverse reaction after getting vaccinated				
Agree	570(79.83%)	144(20.17%)	0.475 *	0.339–0.665
Not sure	528(84.21%)	99(15.79%)	0.640 *	0.449–0.912
Disagree	450(89.29%)	54(10.71%)	Ref	-
Q8. I am afraid of needles				
Agree	168(76.71%)	51(23.29%)	0.466 *	0.326–0.667
Not sure	362(78.02%)	102(21.98%)	0.502 *	0.379–0.665
Disagree	1018(87.61%)	144(12.39%)	Ref	-
Q9. I do not have time to get vaccinated				
Agree	366(84.72%)	66(15.28%)	0.894	0.648–1.233
Not sure	388(79.02%)	103(20.98%)	0.607 *	0.456–0.809
Disagree	794(86.12%)	128(13.88%)	Ref	-
**C4. Perceived benefit of the influenza vaccine**				
Q10. The seasonal influenza jab is effective preventing the seasonal influenza and getting me well				
Disagree	20(60.61%)	13(39.39%)	0.202 *	0.099–0.412
Not sure	140(57.85%)	102(42.15%)	0.180 *	0.134–0.243
Agree	1388(88.41%)	182(11.59%)	Ref	-
Q11. If I get a seasonal influenza jab, it will protect my family member from the seasonal influenza				
Disagree	18(62.07)	11(37.93%)	0.204 *	0.095–0.438
Not sure	148(56.49%)	114(43.51%)	0.162 *	0.121–0.216
Agree	1382(88.93%)	172(11.07%)	Ref	-
Q12. If I get a seasonal influenza jab, it will protect my patients from the seasonal influenza				
Disagree	19(57.58%)	14(42.42%)	0.176 *	0.087–0.356
Not sure	161(60.30%)	106(39.70%)	0.197 *	0.147–0.263
Agree	1368(88.54%)	177(11.46%)	Ref	-
Q13. The seasonal influenza jab is effective reducing the risk of serious clinical outcomes after infection?				
Disagree	21(61.76%)	13(38.24%)	0.208 *	0.102–0.422
Not sure	157(59.25%)	108(40.75%)	0.187 *	0.140–0.250
Agree	1370(88.62%)	176(11.38%)	Ref	-
**C5. Cues to action**				
Q14. Head of department recommended to get seasonal influenza jab?				
Low	88(71.54%)	35(28.46%)	0.341 *	0.224–0.520
Moderate	149(63.95%)	84(36.05%)	0.241 *	0.177–0.328
High	1311(88.05%)	178(11.95%)	Ref	-
Q15. Advertisements for seasonal influenza jab in the hospital?				
Low	50(59.52%)	34(40.48%)	0.189 *	0.119–0.300
Moderate	121(58.45%)	86(41.55%)	0.181	0.132–0.248
High	1377(88.61%)	177(11.39%)	Ref	-
Q16. Family member urge you to get seasonal influenza jab?				
Low	89(66.92%)	44(33.08%)	0.249	0.168–0.370
Moderate	136(60.18%)	90(39.82%)	0.186	0.136–0.254
High	1323(89.03%)	163(10.97%)	Ref	-

^*^ *p*-value < 0.05.

## Data Availability

The data presented in this study are available on request from the corresponding author. To approve a request, it must be justified from a methodological point of view and receive the consent of all authors.

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
