# Peer review of "Exploring Perceptions and Barriers: A Health Belief Model-Based Analysis of Seasonal Influenza Vaccination among High-Risk Healthcare Workers in China"

_vaccines, 2024, doi:10.3390/vaccines12070796_

Round 1

Reviewer 1 Report (Previous Reviewer 1)

Comments and Suggestions for Authors

NIL

Author Response

Thank you for your thorough review and favorable recommendation of our manuscript. We appreciate your time and effort.

Reviewer 2 Report (Previous Reviewer 3)

Comments and Suggestions for Authors

Exploring perceptions and barriers. 2nd rev.

The article is a resubmission. Unfortunately, the system did not make available the authors' response to the previous comment.

1.       The manuscript was revised, but again the authors did not use the journal template.

2.       The authors have correctly given more information on how the data was collected, and the participation rate.

3.       The authors give more information on how the questionnaire was constructed; however, it remains a tool created ad hoc and not validated. At the lines 106 and following, the authors state that “A validity and reliability test has been conducted. Cronbach’s alpha was used to calculate the internal consistency reliability of each construct. Factor analysis, employing varimax rotation, was conducted to validate the questionnaires.” However, these results are not exposed.

4.       In the manuscript the authors confirm the statement "Any questionnaire responses completed within 30 seconds were discarded as they indicated a lack of attention, given that the average time spent on each question was less than 1 second". This statement is self-evident; however, it is not sufficient. Times significantly longer than one second are not sufficient to answer the questions in an online questionnaire. All online collection systems have a time simulator which calculates how many minutes (not seconds!) it takes to complete such a complex questionnaire. The authors should have excluded all questionnaires completed in less than 10 minutes, not 60 seconds.

5.       The authors have improved the discussion, with some reference to other studies.

6.       The fact of having used a self-produced and non-validated questionnaire must be reported in the limitations.

Author Response

Comment 1

 The manuscript was revised, but again the authors did not use the journal template.

Our Response:

Thank you for the reminder. We have utilized the journal template to optimize the formatting.

Comment 2

The authors have correctly given more information on how the data was collected, and the participation rate.

Comment 3

The authors give more information on how the questionnaire was constructed; however, it remains a tool created ad hoc and not validated. At the lines 106 and following, the authors state that “A validity and reliability test has been conducted. Cronbach’s alpha was used to calculate the internal consistency reliability of each construct. Factor analysis, employing varimax rotation, was conducted to validate the questionnaires.” However, these results are not exposed.

Our Response:

Thank you for your suggestion. Regarding the results of the Reliability and Validity analyses, we have presented them in Section 3.2 of the "3. Results" chapter, specifically in lines 161-165.

Additionally, it is true that the questionnaire was self-designed, tailored to our need to convey the content in Chinese and to specifically investigate the preferences of healthcare workers regarding "self-protection, family protection, and patient protection." We have included this clarification as a supplementary note in the "5. Limitation" section, located in lines 313-319.

Comment 4

In the manuscript the authors confirm the statement "Any questionnaire responses completed within 30 seconds were discarded as they indicated a lack of attention, given that the average time spent on each question was less than 1 second". This statement is self-evident; however, it is not sufficient. Times significantly longer than one second are not sufficient to answer the questions in an online questionnaire. All online collection systems have a time simulator which calculates how many minutes (not seconds!) it takes to complete such a complex questionnaire. The authors should have excluded all questionnaires completed in less than 10 minutes, not 60 seconds.

Our Response:

Thank you for your suggestion. In fact, we conducted a small-scale pilot survey prior to the formal experiment to determine a reasonable time for completing the survey questionnaire. The fastest response time recorded was approximately 30 seconds, which is why we chose this duration.

After your thoughtful reminder, we have thoroughly reviewed the response time data for all the questionnaires. Specifically, we found that only 9 individuals (representing 0.49% of the total, the fastest one was 42 seconds) took less than 60 seconds, 353 individuals (19.1%) completed it in less than 120 seconds, a significant majority of 1571 individuals (85.15%) finished within 300 seconds, and 274 individuals (14.85%) took over 300 seconds to complete the questionnaire.

You mentioned that the reason we gave in the main text, "given that the average time spent on each question was less than 1 second," was insufficient to support this choice. We agree with your observation, and therefore, we have revised the justification for choosing 30 seconds to "based on the shortest time of completing the questionnaire from our small-scale pilot experiments before the investigation" in lines 107-108 of the main text.

We have also reviewed and found an article that is similar in topic to ours, which used a questionnaire with 18 questions and a 5-point Likert scale, similar to ours in terms of the number of options. The shortest response time reported in that study was 32 seconds1. We have also added this article to our references.

The issue you mentioned is also very important. We have already reminded readers of this in the 5.limitation section, lines 319-322.

Comment 5

The authors have improved the discussion, with some reference to other studies.

Comment 6

The fact of having used a self-produced and non-validated questionnaire must be reported in the limitations.

Our Response:

Thank you for your suggestion. As we mentioned in our response to Comment 3, it is true that the questionnaire was self-designed, tailored to our need to convey the content in Chinese and to specifically investigate the preferences of healthcare workers regarding "self-protection, family protection, and patient protection." We have included this clarification as a supplementary note in the "5. Limitation" section, located in lines 313-319.

Thank you for these valuable suggestions. We have also made careful modifications, hoping to further guide our research in the future.

  1. Jiang B, Cao Y, Qian J, Jiang M, Huang Q, Sun Y, et al. Healthcare Workers' Attitudes toward Influenza Vaccination: A Behaviour and Social Drivers Survey. Vaccines. 2023; 11:143.

This manuscript is a resubmission of an earlier submission. The following is a list of the peer review reports and author responses from that submission.

Round 1

Reviewer 1 Report

Comments and Suggestions for Authors

Introduction

A meta-analysis revealed that the incidence of seasonal influenza among unvaccinated HCWs was 18.57%, which was 3.4 times higher than among general healthy adults

Comment: Please expand on the above meta-analysis to where when and whom

However, the vaccination rate among HCWs worldwide ranges from 16.9% to 69.4%, which is lower than what is generally expected

Comment: When and published by whom?

HBM model.

Comment: avoid abbreviation without spell it out

Methods

……cluster of seasonal influenza cases in the previous year

Comment: Which year?

Define the HCWs? You have defined them in the results but needed to be in the methods

Comment: what is anticipated sample size? You need to describe how the sample chosen across deferent wards?

Comment: You need to describe the whether the consent form was used or not?

Results

Comment: Could explain how the 65.90%? Response rate obtained and add it in the manuscript

Comment: All the % ie should followed by n=? 85.96% (n=?) were female, and 14.04% (n=?)

Discussions

For instance, a study in the UK reported that 93.8% of healthcare workers received seasonal influenza vaccines13, whereas a study in Germany reported that only

38.8% of physicians were prone to receive the vaccines14. A study in China showed that

the influenza vaccination rate among healthcare workers was 5%, while the vaccination

rate among the elderly aged 60 and above was 0.8%, and the rate among pregnant women

was 4%15.

Comments for all the studies referred could inform when these studies implemented?

Occupation was also an influential factor. Compared to physicians and nurses, surgeons have a significantly lower acceptance rate of influenza vaccines.

Comment: Kindly you need to use studies in relation to occupation as well? The Jordan study was looking at socioecomonic and not Occupational??

These discrepancies highlight the importance of conducting prior surveys to identify specific groups less likely to accept vaccination, allowing infection control practitioners to provide targeted education.

Comment: I don’t see how this study related to the study findings

Conclusion

Comment: you need to summarized the key findings and then followed by the way forward

Reviewer 2 Report

Comments and Suggestions for Authors

Your study highlights the importance of promoting vaccinations among high-risk health workers. This is an important area that needs more intensified work. Here are some areas where I would recommend including either in the methodology, result, or discussion: 

1. Even though a response rate of around 60% is not bad, I recommend taking the needed actions to minimize selection bias. I suggest to compare the main characteristics of those who responded to the survey with those who did not. 

2. Can you please outline the concrete actions you too to promote participation in the survey? Who did it? How?   

3. Did you consider running the survey in different institutions and settings? For example- nursing homes? Please further elaborate on this aspect.

4. I suggest recommending more concrete and practical recommendations on actions to promote vaccination in this group. 

Thanks for this interesting manuscript. 

Reviewer 3 Report

Comments and Suggestions for Authors

The authors conducted a survey to evaluate the reasons why healthcare workers do not accept the influenza vaccine.

1.       Abstract: In this journal the abstract is not structured. Please remove the words "Background, Material and Methods, Results”.

2.       The authors used an ad hoc, non-validated questionnaire. This is a limitation of the study and needs to be justified.

3.       The authors stated that they excluded questionnaires completed in less than 30 seconds. Completing an online questionnaire with 31 questions takes significantly longer than 30 seconds; online platforms generally calculate the expected time. A questionnaire with around 30 questions takes no less than 5 minutes. What's more, the authors tell us that some healthcare workers were unable to respond online and needed to complete a manual questionnaire which was then entered manually. Thirty seconds is really too little.

4.       The authors do not say on what basis they built their questionnaire, if they referred to previous studies on this topic, which are actually very numerous.

5.       In the Discussion, the comparison between the results obtained in this hospital and the many studies on the topic is not developed.

6.       The few references do not give an idea of the vastness of studies in the literature. For example, the prevalence of vaccination acceptance at the beginning of the Discussion should be compared with one of the more than 20 meta-analyses available on this topic. On the acceptance of the influenza vaccination in healthcare workers there are 15 meta-analyses on Pubmed.

7.       In conclusion, the study took into consideration a limited number of topics, which the authors chose at their own discretion, without explaining to readers why. The sample was selected with convenience criteria and had a high response loss. We do not know whether the rate of vaccination acceptance among respondents corresponds to that of vaccination in the hospital. The results have not been included in the international literature. There are no elements that indicate what the research developments are.